# Bioguided Isolation of Alkaloids and Pharmacological Effects of the Total Alkaloid Fraction from *Aspidosperma pyrifolium* Mart. (Apocynaceae)

**DOI:** 10.3390/plants10112526

**Published:** 2021-11-20

**Authors:** Francisca Sabrina Vieira Lins, Vanessa Farias da Silva, Josean Fechine Tavares, Vanda Lúcia dos Santos, Harley da Silva Alves

**Affiliations:** 1Department of Pharmacy, Post-Graduate Program in Pharmaceutical Sciences, State University of Paraíba, Campina Grande 58429-500, PB, Brazil; sabrinav.lins@gmail.com (F.S.V.L.); vandalsantos@servidor.uepb.edu.br (V.L.d.S.); 2National Semiarid Institute, Campina Grande 58429-500, PB, Brazil; belaatiria@hotmail.com; 3Institute for Research in Pharmaceuticals and Medications, Federal University of Paraíba, João Pessoa 58051-900, PB, Brazil; josean@ltf.ufpb.br

**Keywords:** pereiro-preto, plumeran alkaloids, acute toxicity, anti-inflammatory, anti-nociceptive

## Abstract

*Aspidosperma pyrifolium* is used in traditional medicine to treat inflammatory disorders. The aim of the study was to perform phytochemical characterization and evaluate the anti-inflammatory, anti-nociceptive and acute toxicity effects of the total alkaloid fraction (TAF-Ap) from stem barks. Two monoterpenic indole alkaloids were isolated by high performance liquid chromatography coupled with mass spectrometry (HPLC-MS) and the structural elucidation was performed using 1D and 2D NMR analysis. As for toxicity, no animals died at 50 mg kg^−1^ and this concentration presented mild sedation and forced breathing within the first 24 h. The lethal dose capable of killing 50% of the animals (LD_50_) was estimated to be 160 mg kg^−1^. In the pharmacological tests, the models used were 1% carrageenan-induced paw edema and peritonitis, 1% formalin-induced nociception and 1% acetic acid-induced abdominal writhing in Swiss mice. The study made it possible to isolate 15-methoxyaspidospermine and 15-methoxypyrifolidine, corroborating the results of pharmacological assays, which showed anti-inflammatory and analgesic potential, especially at 30 mg kg^−1^ (*p* < 0.001). Thus, the species was shown to be a promising source of active substances, with special attention paid to its toxicological potential.

## 1. Introduction

Inflammation is an immune defense mechanism that the body uses to fight bacteria, viruses and other pathogens [1,2]. In these processes, a variety of chemical mediators are released from damaged tissue, including excitatory amino acids, hydrogen ions, peptides, lipids, and cytokines, all of which underlie inflammation and pain [3].

Non-steroidal anti-inflammatory drugs, such as aspirin, as well as steroidal anti-inflammatory drugs, such as dexamethasone, have been widely used to fight inflammation, but there is clinical evidence that these drugs are capable of causing adverse effects, including gastrointestinal disorders. Alternatively, natural products are growing targets in research for new drug discovery [1,4].

*Aspidosperma pyrifolium* Mart., popularly known as “pereiro-preto”, is a small tree that is widely distributed in northeastern Brazil, more precisely in the Caatinga. Its wood is used in the manufacture of furniture, due to its excellent quality [5,6,7]; in traditional medicine, the extract of its leaves and bark is used for its anti-inflammatory and analgesic properties [7,8,9].

Previous studies have shown a hypotensive effect attributed to the alkaloids present in the bark and leaves of this species, as well as anti-plasmodic activity related to the presence of aspidosperman alkaloids, such as the alkaloid aspidospermin [9,10].

Although there are pharmacological and phytochemical studies of crude extracts from *A. pyrifolium*, there is no information on the therapeutic potential of the isolated fraction of alkaloids present in the species. Therefore, the present study aimed to test the anti-inflammatory and anti-nociceptive potential of the fraction of alkaloids extracted from *A. pyrifolium* husks and to isolate the indolic alkaloids present in this fraction, showing that the therapeutic potential produced is related to these substances.

## 2. Results and Discussion

### 2.1. Chemical Identification of Isolated Compounds

The qualitative analysis by HPLC-DAD of TAF-Ap indicated the presence of two alkaloids. The acquisition of the 3D chromatogram with scanning from 200 to 400 nm (Figure 1) pointed the presence of the alkaloids with absorption greater than 200 mAU and retention times of 7.68 and 7.78 min, respectively.

The mass spectra of **1** and **2** showed molecular ion peaks at *m*/*z* = 385.44 (calcd *m*/*z* 384.241) and *m*/*z* = 415.45 (calcd *m*/*z* 414.252), which allowed us to manage the following structural formulas: C_23_H_32_N_2_O_3_ and C_24_H_34_N_2_O_4_, respectively. The UV spectrum showed three absorption bands at λ_max._ 222, 256 and 285 nm (Appendix A), suggestive of indoline chromophore [11].

The ^1^H NMR data (500 MHz, acetone-d_6_) (Table 1) (Appendix A) showed four signals aromatic hydrogen at δ_H_ 7.16 (t, *J* = 7.8, 1H), δ_H_ 7.06 (d, *J* = 8.2 Hz, 1H), δ_H_ 7.00 (d, *J* = 8.2 Hz, 1H) and δ_H_ 6.82 (dd, *J* = 8.3 and 1.9 Hz, 2H). In addition to these signals, others were visualized between δ_H_ 3.86 and 0.68, suggestive of aliphatic hydrogens [9,11].

The ^13^C-NMR spectrum (125 MHz, acetone-d_6_) (Table 1) (Appendix A) showed 31 signals: 9 from unhydrogenated carbons, 7 from CH carbons, 7 from CH_2_ carbons and 8 from CH_3_ carbons. The signals at δ_C_ 110.38, 110.09, 127.42, 117.39 and 113.45 are characteristic of aromatic methinic carbons. In addition, the signals at δ_C_ 141.65, 141.68, 133.50, 154.66, 150.10 and134.33 are consistent with non-hydrogenated aromatic carbons [9,12].

The heteronuclear correlation map-HSQC (Appendix A) showed the following correlations: δ_H_ 6.82/δ_C_ 110.38 and 110.09 (C-9 and C-11) and δ_H_ 7.16/δ_C_ 127.42 (C-10), suggestive of a tri-substituted aromatic nucleus. The correlations of δ_H_ 7.06/δ_C_ 117.39 (C-9) and δ_H_ 7.00/δ_C_ 113.45 (C-10) are suggestive of another tetra-substituted aromatic ring [9]. The correlations between δ_H_ 3.84/δ_C_ 141.68 e δ_H_ 3.83/δ_C_ 154.66 e δ_H_ 3.86/δ_C_ 150.10, seen in the heteronuclear correlation map-HMBC (Appendix A), confirm the presence of methoxyl groups in C-11, C-11 and C-12 in the two aromatic nuclei, respectively. According to [9], the signals at δ_C_ 169.59 and δ_C_ 170.26 suggested the presence of *N*-acethyl carbonyl groups, in addition to a signal at δ_C_ 52.80 and 52.83, characteristic of C-7 indolic-ring quaternary carbon. These positions were confirmed by long-range heteronuclear correlation at δ_H_ 2.18/δ_C_ 169.59; δ_H_ 2.19/δ_C_ 170.26 and δ_H_ 6.82/δ_C_ 52.80; and δ_H_ 7.06/δ_C_ 52.83.

All the methylene carbons were displayed by ^13^C NMR and experiment APT and the hydrogens attached to them were revealed by chemical shifts, coupling constants and comparison with data from previous research. The signals at δ_C_ 71.17 and 71.15 were attributed to the methinic carbons of the position 21, respectively.

A signal at δ_H_ 3.31 (s, 6H) showed a correlation in the HSQC with δ_C_ 56.74 and suggested the presence of other methoxyl groups in the compounds. Methoxyl insertion at C-15 was reinforced by *α* deprotection in this carbon, *β* deprotection at C-14 and C-20 and *γ* protection at C-19 and C-3, when compared to a structure without this substituent and confirmed by correlations of δ_H_ 3.31/δ_C_ 74.02 and 74.50, seen in the HMBC. In addition, the coupling constants for the two broad doublets at δ_H_ 3.26 (*J* = 9.0 Hz) and 3.17 (*J* = 14.3 Hz) were in accordance with the expected values when the hydrogens H-15 were located in the *α* position in relation to ring *D*. Therefore, it can be deduced that OCH_3_-15 features pseudo-equatorial stereochemistry [13]. The correlation between δ_H_ 2.03/δ_H_ 3.17 and 3.26, observed in the NOESY spectrum (Appendix A), reinforce this argument.

The interpretation of spectral data, in addition to the comparison with the previous research data, made it possible to identify two monoterpenoid indole alkaloids of the plumeran class. Compound **1** was identified as 15-methoxyaspidospermine. The presence of a methoxyl group at position C-11 in compound **2** allowed us to identify it as being t 15-methoxypyrifolidine (Figure 2).

### 2.2. Acute Toxicity

The TAF-Ap dose of 200 mg kg^−1^ was 100% lethal to animals within 30 min of its administration and death was preceded by severe tremors. The 100 mg kg^−1^ dose caused death in a female one hour after administration; by the fourth hour, all the animals showed poor responses to tail pinching, low auricular reflex and impaired posture, as well as tremors and signs of sedation and forced breathing. Regarding the dose of 50 mg kg^−1^, no animal died; however, the animals presented signs of mild sedation and forced breathing in the first 24 h. The lethal dose capable of killing 50% of animals (LD_50_) was estimated to be 160 mg kg^−1^.

The mean values obtained for each group in the weight evolution of the animals showed significant differences for the dose of 100 mg kg^−1^; although they gained weight, the values for body mass increase in the animals of this group were lower than those of the negative control group and of the groups receiving the 50 mg kg^−1^ dose (Table 2).

Regarding the daily feed intake, those animals that received the 100 mg kg^−1^ dose consumed a smaller amount of food when compared to the negative control groups and those receiving the 50 mg kg^−1^ dose (Table 2). Still, as shown in Table 2, the groups that received the 100 mg kg^−1^ dose consumed a larger amount of water than the others. In addition, it was found that there was no significant difference when comparing the water and feed intake of animals receiving the 50 mg kg^−1^ dose in relation to the negative control.

Table 3 shows that there was a significant decrease in the relative weight of the livers of the animals receiving the 100 mg kg^−1^ dose compared to those receiving the 50 mg kg^−1^ dose and the negative control, which corroborates the results of the weight evolution shown in Table 2; these showed a lower weight gain in the animals that received the highest dose.

In addition to body weight changes, the individual weight of each organ can also be considered as an indicator of drug-induced adverse effects, both of which are indicative of toxicity [14]. Regarding the macroscopic characteristics, no alteration in the color or shape of the organs was observed between the studied groups.

The acute toxicity test was useful to establish the doses employed in the anti-inflammatory and anti-nociceptive activity assays. The reason why doses above 50 mg kg^−1^ exerted harmful effects on the animals in the present study is that TAF-Ap is a concentrated fraction of alkaloids, enabling a greater chance of producing toxicity than crude extract.

For this same species, some experimental studies have already been carried out to prove toxicity. In an experimental study of goats at different gestational stages and fed with freshly harvested green leaves from *A. pyrifolium* (4 g kg^−1^ dose), during 19 days of consumption, miscarriage and embryonic losses were observed. Desiccated leaves were not able to cause reproductive changes, although the ingestion of the plant in the first 34 days of gestation was shown to cause mortality [15]. In a survey of 60 respondents residing in the Western and Eastern Seridó of Rio Grande do Norte, including 17 producers and technicians, abortions or the births of weak animals were reported in 16 goats that ingested *A. pyrifolium* leaves during the drought period. Nine respondents also reported the occurrence of poisoning, which was characterized by hind limb stiffness and locomotion difficulty [16].

In a toxicity study of male and female Wistar rats with ethanolic extracts from *A. pyrifolium*, the females demonstrated reduced fetal weight and strong indications of maternal toxicity, in addition to motor disturbances and death at higher concentrations. The male rats were more resistant than the females. In the same study, it was found that *A. pyrifolium* extract promoted hemolysis and was lethal to the *Artemia salina* organism in an in vitro cytotoxicity test [17].

### 2.3. Anti-Inflammatory Activity

#### 2.3.1. Carrageenan-Induced Paw Edema

In this model, an anti-edematogenic effect was observed in both the positive control group (indomethacin) and in the groups treated with TAF-Ap at doses of 20 and c. Here, the animals showed a reduction in the thickness of the paw that received the agent from the first hour until the fourth hour when compared to the negative control (Figure 3).

The 30 mg kg^−1^ dose of TAF-Ap presented a percentage of edema inhibition in the interval between the first and second hour (18.7%) when compared to the group receiving indomethacin (19.5%); after the second hour, the 20 mg kg^−1^ dose also presented a high inhibition percentage (24.5%). The paws of the animals treated with the 30 mg kg^−1^ dose of TAF-Ap and indomethacin as a positive control were shown to present a lower weight at the end of the experiment when compared to the negative control, confirming the anti-edematogenic potential mentioned above (Figure 4).

After tissue injury, blood vessels in the injured area contract momentarily (vasoconstriction), followed by vasodilation and increased blood flow in the area, which can last from 15 min to several hours. The walls of these blood vessels, which normally only allow water and salts to pass through, become permeable, resulting in a rich fluid called exudate that eventually flows to the tissues. This is followed by white cell migration [18].

Paw edema testing is routinely performed to assess the degree of vascular permeability following tissue injury caused by carrageenan injection and promotes biphasic inflammation. This involves the initial phase, which occurs in the first hour after edema induction and features histamine release, 5-hydroxytryptamine, leukotrienes, kinins and cyclooxygenases; and the delayed phase, which runs from the first to the fourth hour, and is related to the production of prostaglandins, bradykinin and neutrophil infiltration [19]. Several published reports suggest that many cytokines, such as TNF-α, IL-1β, IL-2, IL-6 and PGE2, play a role during inflammation. Of these cytokines, TNF-α is the most important actor in inflammatory reactions, generating native protective responses, such as stimulating T cells and macrophages, as well as kinin and leukotriene release, and activating the production of additional inflammatory cytokines. Interleukin-6 (IL-6) is another important cytokine that is released by a variety of cells at the site of injury [18,19,20].

Thus, it is likely that the anti-inflammatory profile of TAF-Ap is related to the activation of this cascade, inhibiting the release of these mediators and, consequently, the vascular events of inflammation, which culminate in the dilation of small arterioles, the extravasation of proteins to the tissue and, consequently, edema formation [18].

#### 2.3.2. Carrageenan-Induced Peritonitis

It was possible to observe a smaller amount of polymorphonuclear cells in the exudates collected in both the indomethacin group (41.0% inhibition) and the 30 mg kg^−1^ TAF-Ap group (43.91% inhibition), showing that there was a significant reduction in the recruitment of inflammatory cells (Figure 5).

An important sign in the inflammation process involves the recruitment of polymorphonuclear cells to the inflammatory foci. Upon stimulation, resident cells produce cytokines to communicate the threat to the injuring agent. These cytokines activate endothelial cells that express leukocyte adhesion molecules, mainly neutrophils; in turn, these produce reactive oxygen species, such as superoxide anion and nitric oxide, which are responsible for causing oxidative stress to eliminate the offending agent [21,22]. When the influx of neutrophils to the inflammatory site is exacerbated, it leads to tissue damage, and can result in serious injury to adjacent tissues when in constant circulation. Therefore, neutrophils play a key role in the pathogenesis of diseases such as atherosclerosis, obesity and rheumatoid arthritis [23,24].

Thus, we can suggest that the anti-inflammatory activity of TAF-Ap may also be related to the modulation of neutrophil migration, as well as the inhibition of the release of inflammatory mediators, as shown in the results observed in the paw edema test.

### 2.4. Anti-Nociceptive Activity

#### 2.4.1. 1% Acetic Acid-Induced Abdominal Writhing Test

Acetic acid is responsible for the secretion of endogenous pain mediators, resulting in the increased synthesis of cyclooxygenases (COX), lipooxygenase (LOX) and prostaglandins, thus stimulating neurons responsible for nociception, which respond to anti-inflammatory drugs [18,25]. In addition, the action of acetic acid on macrophages and basophils in the abdominal cavity induces the release of cytokines such as IL-8, IL-1β and TNF-α, and also triggers vasodilation and vascular permeability [25].

The doses tested showed the inhibition of abdominal writhing: TAF-Ap 10 mg kg^−1^ (75.42%), TAF-Ap 20 mg kg^−1^ (76.67%) and TAF-Ap 30 mg kg^−1^ (84.17%) when compared to the negative control. These data allow us to state that TAF-Ap features analgesic properties (Figure 6). The potent reduction in abdominal contortions observed for TAF-Ap at the doses used may be justified by the inhibition of prostaglandin synthesis, acting on the nociceptive mechanisms of arachidonic acid metabolite processing or release through COX.

#### 2.4.2. Formalin-Induced Nociception Test

The injection of intraplantar formalin into the hind paw of an animal induces severe pain by the direct stimulation of nociceptors, characterized by vigorous licking, bites and bumps on the paw injected with the irritant. This test allows the verification of signals present in the modulation phase of nerve impulses, and also to observe the participation of endogenous systems, such as that of opioids [25,26].

In this model, both the indomethacin group and the 30 mg kg^−1^ TAF-Ap group demonstrated a higher inhibition level than 50%, with less time to stop licking at the end of phase I (neurogenic phase). In phase II (inflammatory phase), all doses of TAF-Ap significantly decreased the time spent licking, with the 30 mg kg^−1^ dose being the most effective (85.57% inhibition) (Table 4).

The results related to the decrease in the number of phase II licks may contribute positively to the anti-inflammatory profile observed in the decrease of carrageenan paw edema, so that both the positive control and the TAF-Ap may have caused the inhibition of mediators responsible for increasing vascular permeability after 30 min, reducing edema caused by the inflammatory agent. The results presented suggest that TAF-Ap exerts a central and peripheral effect against pain and inflammation, which corroborates data from another study [6], in which the 100 mg kg^−1^ dose of the aqueous extract of the seeds from *A. pyrifolium* also showed a decrease in late phase licking time similar to that of positive controls (morphine 5 mg kg^−1^ and indomethacin 10 mg kg^−1^).

It can be inferred that by raising the pain threshold induced by intraperitoneal acetic acid injection and intra-implant formalin injection, and inhibiting the carrageenan-induced edematogenic effect and leukocyte migration, TAF-Ap is endowed with anti-nociceptive and anti-inflammatory activity, both of peripheral origin and associated with central mechanisms of pain and inflammation inhibition. These activities are related to the indolic alkaloids found in the extracts of the species of genus *Aspidosperma*, which could be confirmed from the isolation of the compounds 15-methoxyaspidospermine and 15-methoxypyrifolidine. This was the first pharmacological study conducted with the isolated fraction of alkaloids of the species *A. pyrifolium.*

## 3. Materials and Methods

### 3.1. Vegetable Material Collection and Identification

The stem barks from *A. pyrifolium* Mart. were collected at Capim Grande site, São José da Mata district, Campina Grande, Paraíba, in August 2014 (coordinates: S 7°13′27.56″–W 36°00′53.37″). After collection, the material was identified by Prof. Dr. Leonardo Pessoa Felix and the exsiccata deposited in the Herbarium Jayme Coelho de Moraes at the Federal University of Paraíba, under the number 20104.

### 3.2. Isolation and Characterization of Compounds

The stem barks from *A. pyrifolium* Mart. were dried in an air circulation oven (40 °C) and powdered in a 10 mesh knife mill. The powder was exhaustively extracted with 96% (*v*/*v*) ethanol and rotary evaporated (Tecnal TE-21) to yield 552 g of crude ethanolic extract (CEE). Next, 112 g of CEE was subjected to a liquid-liquid partition, providing the following phases: hexane (57.46%/64.36 g), chloroform (18.03%/20.2 g), ethyl acetate (1.25%/1.4 g) and methanol: H2O—7:3 (*v*/*v*) (3.83%/4.28 g).

### 3.3. Obtaining of the Total Alkaloids Fraction from A. pyrifolium (TAF-Ap)

The CEE (110 g) was submitted for extraction of the total alkaloids fraction. The extract was initially treated with 2 L of 3% hydrochloric acid solution and subsequently filtered on filter paper. The residue was discarded and the filtrate was subjected to several extractions with chloroform. The acidic aqueous phase was basified with NH_4_OH, stirred vigorously to pH 10, and then extracted with chloroform [27]. Subsequently, the chloroform fraction was evaporated under reduced pressure in a rotary evaporator at 40 °C, yielding 2.06 g of the TAF-Ap.

#### Isolation of Alkaloids from TAF-Ap

The TAF-Ap (1.90 g) was subjected to medium pressure chromatography (BUCHI Pump Manager, model C-615, Flawil, Switzerland) using aluminum oxide 90 (70-270 mesh ASTM, MN) and binary mixtures of hexane and methanol in an increasing polarity gradient, providing 48 fractions. The chromatographic profile of sample 8 (126 mg) was analyzed in an analytical HPLC (Infinity Series More Confident 1260 system, Agilent, Santa Clara, CA, United States). The fraction was diluted in 0.2% trifluoroacetic acid (TFA):acetonitrile (ACN) (1:1—*v*/*v*) aqueous solution at 1 mg mL^−1^ and the volume injected into the apparatus was 50 μL. A reverse phase C-18 column was maintained at 40 °C and 0.2% TFA:ACN aqueous solution (77:23 to 65:35 *v*/*v*) was used as a mobile phase with a flow of 10 mL min^−1^, a time of 15 min, and in gradient mode. After analysis in analytical HPLC, the sample was injected into semi-preparative HPLC (2767 Sample Manager System, Waters, with a UV-Vis photodiode detector, Milford, Massachusetts) using the same conditions as previously mentioned. At the end of the separation, a total of 10 fractions were obtained (Ap-1 to Ap-10). Fraction Ap-3 was directed to structural analysis, which allowed the identification of compounds 1 and 2 (Figure 2).

### 3.4. Techniques Used for Structural Identification

The mass spectra were obtained by injecting the samples into a high-performance liquid chromatograph with a diode array detector coupled with a low resolution electrospray ionization mass spectrometer (HPLC-DAD-ESI-MS; 2767 Sample Manager System, Waters, Milford, Massachusetts) and a quadrupolar analyzer ion trap (trap ions) operating in the positive mode. The ^1^H and ^13^C NMR spectra (500 and 125 MHz, acetone-d_6_) (1D and 2D) were recorded on a Varian Mercury spectrometer using TMS as the internal standard.

### 3.5. Animals

The study was carried out in strict accordance with the Standard Operating Procedures (Laboratory of Pharmacology at the State University of Paraiba, Campina Grande, Brazil) and approved by a veterinarian, who frequently monitored the health of the animals through physical condition assessments. All efforts were made to reduce the suffering of the experimental animals.

The animals were obtained from the Institute for Research in Pharmaceuticals and Medications (Iperfarm) of the Federal University of Paraíba (UFPB). Disease-free adult mice Swiss (*Mus musculus*) of both sexes, weighing between 25 and 30 g, were used. The animals were housed in standard plastic cages in an environment with a controlled temperature and humidity, a light-dark cycle of 12 h and food and water ad libitum. All the mice underwent a period of at least 7 days of acclimatization prior to the procedure, being socialized through contact, including with humans. The animals were handled with care to minimize stress. The researchers confirm that the laboratory established a protocol for the use of humane endpoints in cases where animals became severely ill or moribund during the experiment, but none reported death or behavioral changes in animals. The animals received standard laboratory pellets and water ad libitum for both the adaptation period (7 days) and during the trial, except for the period of 12 h prior to the experiments, in which the access to food was restricted. Throughout the experiments, all of the animals received humane care according to the ‘‘Guide for the Care and Use of Laboratory Animals” prepared by the National Academy of Sciences [28]. At the end of the experiments, the animals were euthanized by intraperitoneal anesthetic overdose (ketamine 50 mg kg^−1^ and xylazine—3 mg kg^−1^), following the recommendation of resolution number 714 of 20 June 2002 of the Federal Council of Veterinary Medicine. The experimental protocols were submitted and approved by the Ethics Committee on Animal Use (CEUA) of the Faculty of Social and Applied Sciences of Campina Grande (FACISA)—Project Number/Protocol: 5402042015 and CIAEP/CONCEA: 01.001.2012.

### 3.6. Chemicals and Reagents

The following drugs and chemicals were used: indomethacin 10 mg (Bayer, Leverkusen, Germany), dipyrone sodium 500 mg (TEUTO, Goiás, Brazil), carrageenan (laboratory (BDH Chemicals^®^, London, UK), formalin (BDH Chemicals^®^, London, UK), (Merck, Kenilworth, NJ, USA), HPLC-grade acetonitrile (Merck, Germany, USA) and trifluoroacetic acid (Merck, Kenilworth, NJ, USA). TAF-Ap was dissolved in 0.9% saline solution. All the solutions were prepared immediately prior to the start of the experiments.

### 3.7. Pharmacological Tests

#### Acute Toxicity Assessment

The acute toxicity test was based on ANVISA Resolution No. 90/2004 and the Almeida Experimental Protocol [29], with some adaptations according to [30], where the behavioral alterations in the central and autonomic nervous system were evaluated along with the occurrence of death [31]. In this trial, a total of 48 adult mice (*n* = 5), corresponding to 8 groups, with 24 males and 24 females, were treated orally at doses of 50, 100 and 200 mg kg^−1^ of TAF-Ap; a control group received saline solution only. Parameters such as body mass, water and feed intake and excreta production were evaluated for 30 min, 1, 2, 4, and every 24 h for 14 days. On day 15, the animals were weighed and euthanized. After euthanasia, weighing and macroscopic analysis of the viscera (liver, kidneys, spleen, lungs and heart) was carried out [32].

### 3.8. Evaluation of Anti-Inflammatory Activity

#### 3.8.1. Carrageenan-Induced Paw Edema

The test was based on the methodology used by Zayed and Hassan (2014) [33]. Five groups of six animals each (*n* = 6) were treated orally with saline (0.9%; negative control), 10 mg kg^−1^ indomethacin (positive control) and TAF-Ap at doses of 10, 20 and 30 mg kg^−1^. After 30 min, 0.1 mL of 1% carrageenan solution was injected into the right hind paw subplantar region. Paw thickness was measured with a digital caliper before and up to the fourth hour after edematogenic stimulation. The anti-edematogenic effect was obtained by calculating the difference between the initial thickness of the paw that received the phlogistic agent and the thickness measured after each hour. The inhibition percentage was calculated using the following formula: Inhibition = (V − X)/V × 100%, where V is the measure of vehicle group edema and X tests or positive control. To complement the results, the paw that received the phlogistic agent was cut with surgical scissors in the tibio-tarsal region for later comparison between weights, according to the methodology used by [34].

#### 3.8.2. Carrageenan-Induced Peritonitis

Four groups of six animals each (*n* = 6) were treated orally with 10 mg kg^−1^ saline (negative control), 10 mg kg^−1^ indomethacin and TAF-Ap at doses of 20 and 30 mg kg^−1^. After 30 min, a 1% carrageenan solution (0.1 mL/10 g) was injected into the intraperitoneal cavity of the animals. Four hours after the induction of inflammation, the animals were sacrificed and the intraperitoneal cavity washed with 2 mL of alkaline phosphate buffer (pH 7.2). The solution containing the buffer plus the peritoneum cells was transferred to tubes containing 0.4 mL of Turk’s solution. After five minutes, the global polymorphonuclear leukocyte count was performed in the Neubauer Chamber [35].

### 3.9. Evaluation of Anti-Nociceptive Activity

#### 3.9.1. Acetic Acid-Induced Abdominal Writhing Test

Five groups of six animals each (*n* = 6) were treated orally with 10 mg kg^−1^ saline (negative control), dipyrone 500 mg kg^−1^ (positive control) and TAF-Ap at doses of 10, 20 and 30 mg kg^−1^. After 40 min, a 1% acetic acid solution (0.1 mL 10 g^−1^) was injected into the intraperitoneal cavity and the animals were transferred to clear glass funnels and observed for 20 min to account for the number of abdominal contortions, followed by stretching of the lower limbs [36].

#### 3.9.2. Formalin-Induced Nociception

Five groups with six animals each (*n* = 6) were treated orally with 10 mg kg^−1^ saline (negative control), 10 mg kg^−1^ indomethacin (positive control) and TAF-Ap at doses of G10, 20 and 30 mg kg^−1^. After 30 min, 20 µL of 1% formalin was injected into the subplantar region of the right paw of the animals. The animals’ reactivity was observed by recording the start and end time of licking the paw that received the phlogistic agent during the initial 5 min (Phase I—neurogenic) and between 15 and 30 min (Phase II—inflammatory). The calculation involved subtracting the final time by the initial time in each phase (Tf—To) [37].

### 3.10. Statistical Analysis

The results were evaluated using analysis of variance (ANOVA) followed by Tukey’s post-test. All the results were expressed as mean ± standard deviation (s.d.) with a minimum significance level of *p* < 0.05 and analyzed using GraphPad Prism 5.0 software.

## 4. Conclusions

The results obtained in this research showed that *Aspidosperma pyrifolium* is an important plant species for the research of bioactive compounds. In this study, two monoterpenic indole alkaloids were isolated from the total alkaloid fraction. Through spectroscopic techniques, it was possible to identify 15-methoxyaspidospermine and 15-methoxypyrifolidine, compounds previously found in the species.

In the acute toxicity test, it was demonstrated that the TAF-Ap has considerable toxicity under the conditions evaluated and the LD_50_ was estimated at 160 mg kg^−1^. However, a potent anti-inflammatory effect was demonstrated in the paw edema and peritonitis tests induced by carrageenan, especially at the dose of 30 mg kg^−1^ (*p* < 0.001) of TAF-Ap, promoted by the inhibition of the release of inflammation mediators and the modulation of neutrophil migration, respectively. In addition, TAF-Ap also showed interesting analgesic properties in the acetic acid-induced nociception test, in which there was an 84% decrease in the number of abdominal contortions at the dose of 30 mg kg^−1^, when compared to the negative control; and in the formalin test, with a significant decrease in the number of licks on the paw where the phlogistic agent was applied. Thus, the species whas shown to be a promising source of substances with pharmacological activity, with special attention paid to the toxicological potential presented. Therefore, its use should be conditioned to the determination of safe doses.

## Figures and Tables

**Figure 1 plants-10-02526-f001:**
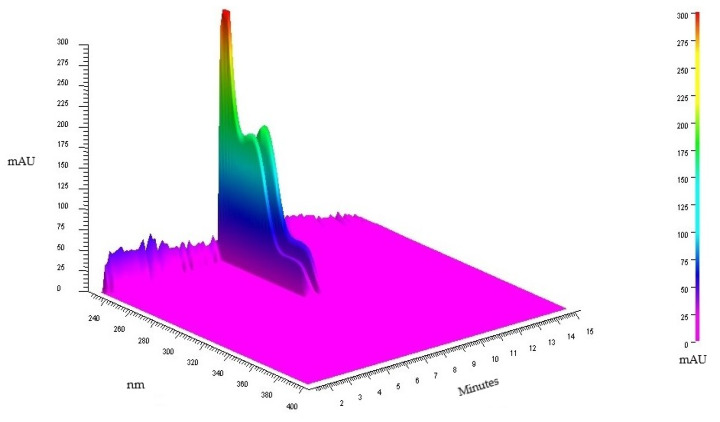
3D chromatogram of an exploratory analysis of TAF-Ap.

**Figure 2 plants-10-02526-f002:**
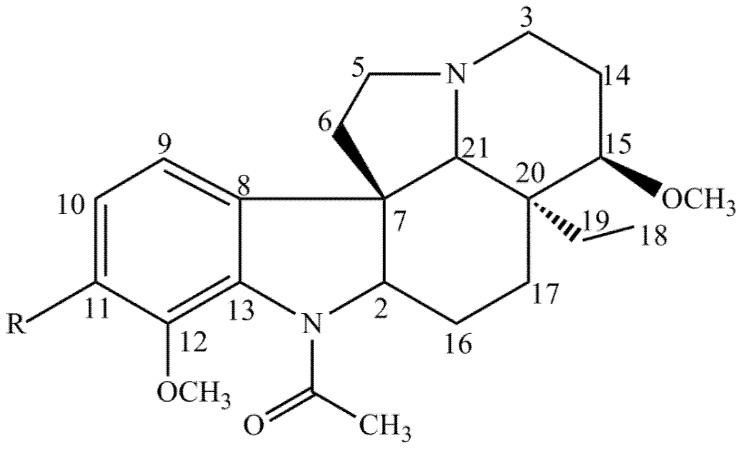
Compounds isolated from *A. pyrifolium* present in TAF-Ap. **1**—R=H, **2**—R=OCH_3_.

**Figure 3 plants-10-02526-f003:**
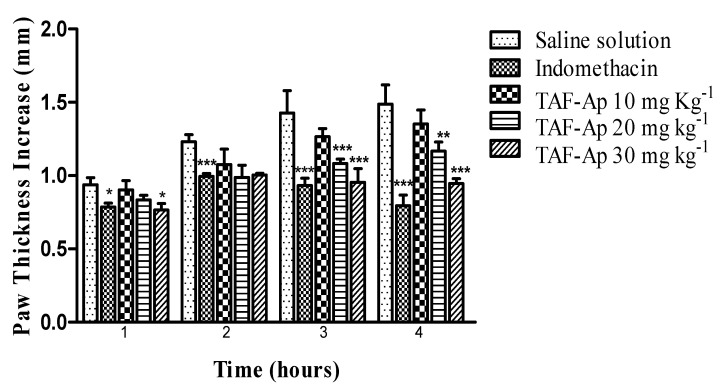
Evaluation of the anti-edematogenic effect of the total alkaloid fraction from *A. pyrifolium* (TAF-Ap) as a function of time after the induction of 1% carrageenan paw edema. Analysis of variance (ANOVA) followed by Tukey’s post-test, * *p* < 0.05; ** *p* < 0.01; *** *p* < 0.001. The absence of asterisks shows that the data were not significant.

**Figure 4 plants-10-02526-f004:**
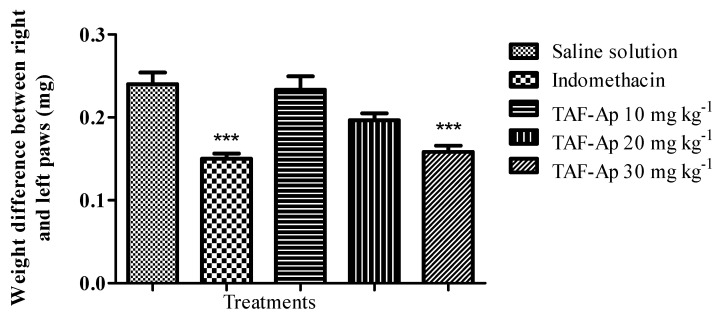
Comparison of paw weights that received 1% carrageenan after the 4 h period that marked the end of the experiment. Analysis of variance (ANOVA) followed by Tukey’s post-test (*** *p* < 0.001). The absence of asterisks shows that the data were not significant.

**Figure 5 plants-10-02526-f005:**
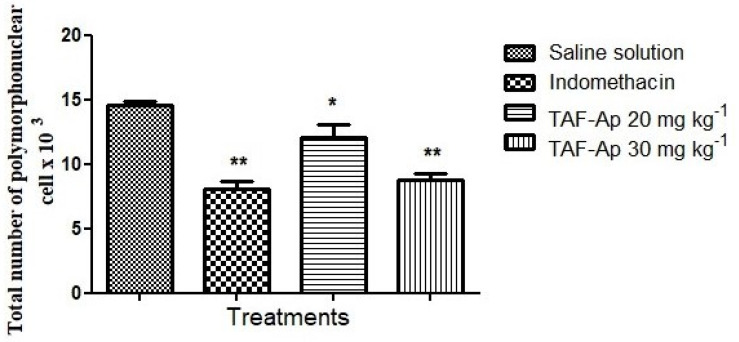
Comparison of the total number of polymorphonuclear cells observed in the intraperitoneal cavity of animals after 4 h of inflammation induction by 1% carrageenan solution. Analysis of variance (ANOVA) followed by Tukey’s post-test * *p* < 0.05, ** *p* < 0.01.

**Figure 6 plants-10-02526-f006:**
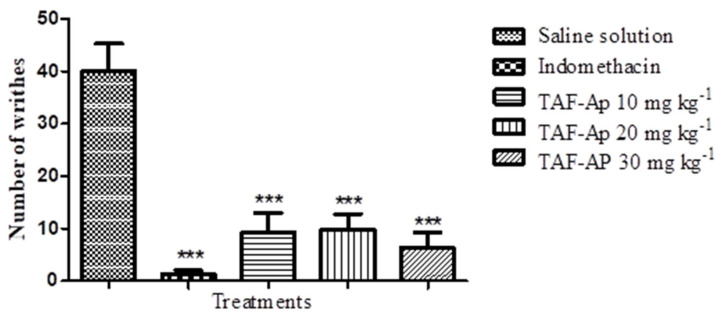
Evaluation of the effect of the total alkaloid fraction from *A. pyrifolium* (TAF-Ap) on the decrease in the number of abdominal writhing induced by the intraperitoneal injection of 1% acetic acid. Analysis of variance (ANOVA) followed by Tukey’s post-test (*** *p* < 0.001).

**Table 1 plants-10-02526-t001:** ^1^H and ^13^C NMR data of 15-methoxyaspidospermine (**1**) and 15-methoxypyrifolidine (**2**) (δ, acetone-d_6_, 500 and 125 MHz).

Position	1		2	
C	δ_C_	δ_H_	δ_C_	δ_H_
**2**	67.67	4.93 (dd, *J =* 6.2 and 10.7 Hz, 1H)	67.90	4.85 (dd, *J =* 6.1 and 10.6 Hz, 1H)
**3**	52.83	3.26 (brd, *J* = 8.2 Hz, 1H) and 2.17 (m, 1H)	52.70	3.26 (brd, *J* = 8.2 Hz, 1H) and 2.17 (m, 1H)
**5**	52.80	3.37 (d, *J* = 6.8 Hz, 1H) and 2.36 (m, 1H)	52.70	3.37 (d, *J* = 6.8 Hz, 1H) and 2.36 (m, 1H)
**6**	37.71	2.06 (brt, *J* = 2.2 Hz, 1H) and 1.92 (dd, *J* = 3.8 and 15.2 Hz, 1H)	37.35	(brt, *J* = 2.2 Hz, 1H) and 1.92 (dd, *J* = 3.8 and 15.2 Hz, 1H)
**7**	52.80	-	52.83	-
**8**	141.65	-	141.65	-
**9**	110.38	6.82 (dd, *J* = 8.3 and 1.9 Hz, 2H)	117.39	7.06 (d, *J* = 8.2 Hz, 1H)
**10**	127.42	7.16 (t, *J* = 7.8, 1H)	113.45	7.00 (d, *J* = 8.2 Hz, 1H)
**11**	110.09	6.82 (dd, *J* = 8.3 and 1.9 Hz, 2H)	154.66	-
**12**	141.68	-	150.10	-
**13**	133.50	-	134.33	-
**14**	24.44	2.03 (m, 1H) and 1.59 (brd, *J* = 3.8 Hz, 1H)	24.33	2.03 (m, 1H) and 1.59 (brd, *J* = 4.5 Hz, 1H)
**15**	74.02	3.26 (brd, *J =* 9.0 Hz, 1H)	74.50	3.17 (brd, *J =* 14.3 Hz, 1H)
**16**	24.77	2.00 (m, 1H) and 1.35 (m, 1H)	24.60	2.00 (m, 1H) and 1.35 (m, 1H)
**17**	24.33	2.03 (m, 1H) and 1.37 (m, 1H)	24.44	2.03 (m, 1H) and 1.37 (m, 1H)
**18**	6.77	0.69 (t, *J* = 7.5 Hz, 3H)	6.84	0.68 (t, *J* = 7.5 Hz, 3H)
**19**	30.45	1.05 (q, *J* = 7.3 Hz, 2H)	29.98	1.00 (q, *J* = 7.3 Hz, 2H)
**20**	36.59	-	36.57	-
**21**	71.17	3.80 (s, 1H)	71.15	3.81 (s, 1H)
**11-OCH_3_**	-	-	56.53	3.83 (s, 3H)
**12-OCH_3_**	56.01	3.84 (s, 3H)	56.53	3.86 (s, 3H)
**15-OCH_3_**	56.74	3.31 (s, 3H)	56.74	3.31 (s, 3H)
**NCOCH_3_**	169.59		170.26	
**NCOCH_3_**	22.93	2.18 (s, 3H)	22.98	2.19 (s, 3H)

**Table 2 plants-10-02526-t002:** Effect of oral administration of the total alkaloid fraction from *A. pyrifolium* (TAF-Ap) on weight evolution (change in body mass) by water and feed intake in 14 day-old male and female Swiss mice.

Parameter	Sex	Saline Solution	TAF-Ap (50 mg kg^−1^)	TAF-Ap (100 mg kg^−1^)
**Initial W. (g)**	**M**	28.83 ± 1.32	27.67 ± 2.58	30.67 ± 0.81
**Final W. (g)**		33.17 ± 1.31	34.50 ± 3.72	31.50 ± 3.72
**Gain (%)**		4.34	6.83	0.83 ***
**Initial W. (g)**	**F**	27.50 ± 0.83	25.00 ± 1.41	25.33 ± 3.14
**Final W. (g)**		32.00 ± 1.26	32.17 ± 1.54	26.31 ± 3.18
**Gain (%)**		4.50	7.17	0.98 ***
**Feed intake per day (g)**	**M**	34.57 ± 2.92	37.29 ± 3.42	25.50 ± 3.25 ***
	**F**	36.71 ± 2.94	35.86 ± 2.65	23.07 ± 2.30 ***
**Water consumption per day (mL)**	**M**	52.14 ± 4.25	52.86 ± 4.68	56.43 ± 2.25 ***
	**F**	46.79 ± 4.64	50.00 ± 3.39 ***	60.29 ± 3.93 ***

Initial W. = initial weight; Final W. = final weight; M = males; F = females. Results are expressed as mean ± sd. (*n* = 5). Analysis of variance (ANOVA) followed by Tukey’s post-test, *** *p* < 0.001. The absence of asterisks shows that the data were not significant.

**Table 3 plants-10-02526-t003:** Effect of oral administration of the total alkaloid fraction from *A. pyrifolium* (TAF-Ap) on relative organ weight of male and female Swiss mice at the end of the experiment.

Relative Weight of Organs (g/100 g)	Sex	Negative Control	TAF-Ap (50 mg kg^−1^)	TAF-Ap (100 mg kg^−1^)
**Liver**	**M**	5.11 ± 0.65	5.06 ± 0.55	4.13 ± 0.54 *
**Spleen**		0.59 ± 0.32	0.73 ± 0.47	0.80 ± 0.21
**Heart**		0.47 ± 0.02	0.50 ± 0.05	0.52 ± 0.11
**Kidneys**		1.23 ± 0.64	1.37 ± 0.11	1.24 ± 0.16
**Liver**	**F**	4.98 ± 0.20	4.95 ± 0.47	4.03 ± 0.53 *
**Spleen**		0.59 ± 0.11	0.45 ± 0.11	0.48 ± 0.25
**Heart**		0.48 ± 0.03	0.45 ± 0.04	0.36 ± 0.18
**Kidneys**		1.08 ± 0.11	0.97 ± 0.08	0.85 ± 0.44

M = males; F = females. Results are expressed as mean ± sd. (*n* = 5). One-way analysis of variance (ANOVA) followed by Tukey’s post-test, * *p* < 0.05. The absence of asterisks shows that the data were not significant.

**Table 4 plants-10-02526-t004:** Antinociceptive effect of the total alkaloid fraction from *A. pyrifolium* (TAF-Ap) in phases I (0–5 min) and II (15–30 min) after 1% formalin-induced nociception.

Treatments	Time Animals Are Licking Paw in Phase I (0–5 min)	Inhibition (%)	Time Animals Remain Licking Paw in Phase II (15–30 min)	Inhibition (%)
**Saline**	4.53 ± 0.34		11.30 ± 1.66	
**Indomethacin**	1.58 ± 1.39 **	62.12 **	1.00 ± 1.43 ***	91.15 ***
**TAF-Ap 10 mg kg^−1^**	3.14 ± 1.09	30.68	6.90 ± 1.5	38.93
**TAF-Ap 20 mg kg^−1^**	2.98 ± 1.00	34.21	3.4 ± 2.0 **	69.91 **
**TAF-Ap 30 mg kg^−1^**	1.90 ± 1.09 **	58.05 **	1.63 ± 1.79 ***	85.57 ***

Results are expressed as mean ± sd. (*n* = 6). Analysis of variance (ANOVA) followed by Tukey’s post-test, ** *p* < 0.01; *** *p* < 0.001. The absence of asterisks shows that the data was not significant.

## Data Availability

Data available on request.

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
