# Peer review of "Bioguided Isolation of Alkaloids and Pharmacological Effects of the Total Alkaloid Fraction from Aspidosperma pyrifolium Mart. (Apocynaceae)"

_plants, 2021, doi:10.3390/plants10112526_

Round 1
Reviewer 1 Report
- The manuscript is well written and presents interesting results. It aimed to test the anti inflammatory and anti-nociceptive potential of the fraction of alkaloids extracted from A. pyrifolium husks and to isolate indolic alkaloids from this fraction. This with the goal to demonstrate that the therapeutic potential produced is related to these substances.
- The abstract mentions isolation of compound 1 and compound 2. This is obscure. Specific information about the compounds should be included.
- The manuscript should be revised for accuracy of writing of chemical formulas. For example, in line 318 NH4OH should be revised, 4 should be in subscript.
- Line 301: The sentence should be revised " at Capim Grande site".
- Line 308: is sprayed the correct word? Or should it be ground/crushed?
- Line 314: you should not start the sentence with the word -here. The sentence should be revised.
- Similarly, in line 322' the sentence should not be started with the word – Next. The sentence should be revised.
- Section 3.3 . If this is not an original biochemistry assay procedure, then a reference should be added to the method used.
- Line 379: should ANVISA RESOLUTION be written in capital letters?
- Conclusion: the last sentence of the conclusion does not belong to this section: " s. In addition, other studies should be conducted to measure the mechanisms of action involved in therapeutic potential, focusing on isolated substances". It is not a conclusion of the conducted study.
- Figure 1: the figure should be improved. It is impossible to see the titles of the axes, they are to small, and the numbers should also be enlarged for clarity if possible.
- Line 65-75, and text throughout the results section: This text is fragmented to too many small sections, sometimes containing only one or two sentences. An attempt should be made to combine the text into more substantial paragraph. I.e., the text itself does not needs to be changed, just avoid sectioning it into such small sections.
- Figure 2: It is not clear what is the chemical form in the figure? The titles say compounds (in plural) isolated..
- Table 2: in the column headings the units should be written in parentheses. Also, do not use abbreviation in the title of the figure. Furthermore, the heading of the figure should be improved to include the information that both M and F data are presented.
- Same comments apply to Table 3 as well.
- Figure 3: It is not clear to comparison between what are the asterisk referring to? They are located weirdly on the lines. Also, avoid abbreviations in the figure legend. TAF-Ap a. The legend should be stand alone (this should be checked for all figure legends in the manuscript).
- Figure 4, Figure 5, and figure 6: Again, it is not clear what was analyzed statistically. Asterisks represent significance difference of what from what?
Author Response
Article
Bioguided isolation of alkaloids and pharmacological effects of the total alkaloid fraction from Aspidosperma pyrifolium Mart. (APOCYNACEAE)
- The manuscript is well written and presents interesting results. It aimed to test the anti inflammatory and anti-nociceptive potential of the fraction of alkaloids extracted from A. pyrifolium husks and to isolate indolic alkaloids from this fraction. This with the goal to demonstrate that the therapeutic potential produced is related to these substances.
We appreciate your attention and suggested corrections to improve our work.
2. The abstract mentions isolation of compound 1 and compound 2. This is obscure. Specific information about the compounds should be included.
We made the correction
3. The manuscript should be revised for accuracy of writing of chemical formulas. For example, in line 318 NH4OH should be revised, 4 should be in subscript.
We made the correction
4. Line 301: The sentence should be revised " at Capim Grande site".
We made the correction
5. Line 308: is sprayed the correct word? Or should it be ground/crushed?
We made the word
6. Line 314: you should not start the sentence with the word -here. The sentence should be revised.
We made the correction
7. Similarly, in line 322' the sentence should not be started with the word – Next. The sentence should be revised.
We made the correction
8. Section 3.3. If this is not an original biochemistry assay procedure, then a reference should be added to the method used.
We added a reference
9. Line 379: should ANVISA RESOLUTION be written in capital letters?
Only the term ANVISA which is an acronym
10. Conclusion: the last sentence of the conclusion does not belong to this section: " s. In addition, other studies should be conducted to measure the mechanisms of action involved in therapeutic potential, focusing on isolated substances". It is not a conclusion of the conducted study.
We removed the excerpt that was not part of the conclusion. In addition, we redid a good part of the conclusion including information relevant to the research.
11. Figure 1: the figure should be improved. It is impossible to see the titles of the axes, they are to small, and the numbers should also be enlarged for clarity if possible.
We've increased the axis titles and figure size to preview the numbers.
12. Line 65-75, and text throughout the results section: This text is fragmented to too many small sections, sometimes containing only one or two sentences. An attempt should be made to combine the text into more substantial paragraph. I.e., the text itself does not needs to be changed, just avoid sectioning it into such small sections.
We joined the paragraphs that had the same sequence of ideas
13. Figure 2: It is not clear what is the chemical form in the figure? The titles say compounds (in plural) isolated.
We put in the radical information that shows the two chemical compounds
14. Table 2: in the column headings the units should be written in parentheses. Also, do not use abbreviation in the title of the figure. Furthermore, the heading of the figure should be improved to include the information that both M and F data are presented.
We accept the suggestions and improve the title
15. Same comments apply to Table 3 as well.
We accept the suggestions and improve the title
16. Figure 3: It is not clear to comparison between what are the asterisk referring to? They are located weirdly on the lines. Also, avoid abbreviations in the figure legend. TAF-Ap a. The legend should be stand alone (this should be checked for all figure legends in the manuscript).
We have redone the figure in the form of bars for ease of interpretation. In addition, we placed the meaning of the TAF-Ap code in the figure title and made it very clear in the article what it is about.
17. Figure 4, Figure 5, and figure 6: Again, it is not clear what was analyzed statistically. Asterisks represent significance difference of what from what?
Indeed, the titles of the figures were confused and were redone. The asterisks show that there was a statistical difference between the doses of TAF-Ap tested with the negative control.

Reviewer 2 Report
Article
Bioguided isolation of alkaloids and pharmacological effects of the total alkaloid fraction from Aspidosperma pyrifolium Mart. (APOCYNACEAE)
A brief summary
In this work the alkaloids fraction (containing two alkaloids identyfied as 15-meth-oxyaspidospermine and 15-methoxypyrifolidine) isolated form Aspidosperma pyrifolium stem barks, traditionally used to treat inflammatory disorders, the anti-inflammatory, anti-nociceptive and acute toxicity effects was evaluated. The paper is very interesting and valuable. The research is well designed, performed, analysed and described although it needs some improvements.
Broad comments
1. A glossary of abbreviations used in the text would be very useful (eg. CEE-Ap, TAF-Ap). Is TAF the same as TAF-Ap?
2. The text contains numerous typos and transcription errors (eg. APOCYNACEAE, 40 ºC, NH4OH, 40°C).
3. Will additional material be available to readers? If not, they should not be referred to in the text (eg. lines 68, 83, 104).
4. Key words need improvement.
Specific comments
Line 56 – “Chemicals of isolated compounds” – please explain.
Figure 1. The “In the image, we can see the two alkaloids” description is perhaps too literal. On the other hand, the 3D view presented does not allow to see if the separation of the two compounds mentioned is complete (up to the baseline). A description of the separation parameters here is unnecessary - this can be found in the Materials and Methods section.
Figure 2. The figure shows only one molecule with an unsigned substituent. The description should be completed as follows: “Compounds isolated from A. pyrifolium. present in TAF”.
Author Response
Article
Bioguided isolation of alkaloids and pharmacological effects of the total alkaloid fraction from Aspidosperma pyrifolium Mart. (APOCYNACEAE)
A brief summary
In this work the alkaloids fraction (containing two alkaloids identyfied as 15-meth-oxyaspidospermine and 15-methoxypyrifolidine) isolated form Aspidosperma pyrifolium stem barks, traditionally used to treat inflammatory disorders, the anti-inflammatory, anti-nociceptive and acute toxicity effects was evaluated. The paper is very interesting and valuable. The research is well designed, performed, analysed and described although it needs some improvements.
We are pleased with the compliments and appreciate the suggestions
Broad comments
- A glossary of abbreviations used in the text would be very useful (eg. CEE-Ap, TAF-Ap). Is TAF the same as TAF-Ap?
We placed the definitions of acronyms in the text itself and standardized what left doubts.
- The text contains numerous typos and transcription errors (eg. APOCYNACEAE, 40 ºC, NH4OH, 40°C).
We fix the errors
- Will additional material be available to readers? If not, they should not be referred to in the text (eg. lines 68, 83, 104).
We make the spectra available as supplementary material.
- Key words need improvement.
We exchanged some keywords and improved others.
Specific comments
Line 56 – “Chemicals of isolated compounds” – please explain.
We corrected this title so as not to generate doubts.
Figure 1. The “In the image, we can see the two alkaloids” description is perhaps too literal. On the other hand, the 3D view presented does not allow to see if the separation of the two compounds mentioned is complete (up to the baseline). A description of the separation parameters here is unnecessary - this can be found in the Materials and Methods section.
The description of the alkaloid separation parameters will only be shown in materials and methods
Figure 2. The figure shows only one molecule with an unsigned substituent. The description should be completed as follows: “Compounds isolated from A. pyrifolium. present in TAF”.
We put the two substituent and complete the title.
